# External Validation of a Radiomics Model for the Prediction of Complete Response to Neoadjuvant Chemoradiotherapy in Rectal Cancer

**DOI:** 10.3390/cancers14041079

**Published:** 2022-02-21

**Authors:** Anaïs Bordron, Emmanuel Rio, Bogdan Badic, Omar Miranda, Olivier Pradier, Mathieu Hatt, Dimitris Visvikis, François Lucia, Ulrike Schick, Vincent Bourbonne

**Affiliations:** 1Radiation Oncology Department, University Hospital, 29200 Brest, France; anais.bordron@chu-brest.fr (A.B.); omar.miranda@chu-brest.fr (O.M.); olivier.pradier@chu-brest.fr (O.P.); francois.lucia@chu-brest.fr (F.L.); ulrike.schick@chu-brest.fr (U.S.); 2Radiation Oncology Department, ICO, 44800 Nantes, France; emmanuel.rio@ico.unicancer.fr; 3LaTIM, UMR 1101 INSERM, Bretagne Occidentale University, 29200 Brest, France; bogdan.badic@chu-brest.fr (B.B.); hatt@univ-brest.fr (M.H.); dimitris@univ-brest.fr (D.V.); 4Department of General and Digestive Surgery, University Hospital, 29200 Brest, France; 5Radiation Oncology Department, CHIC Quimper, 29000 Quimper, France

**Keywords:** radiotherapy, chemotherapy, colorectal cancer, colorectal surgery, magnetic resonance imaging

## Abstract

**Simple Summary:**

In locally advanced rectal cancer (LARC), a minority of patients presents a pathological complete response (pCR) after neoadjuvant chemoradiotherapy (CRT). In this sub-population, organ preservation could be proposed without compromising overall survival. Using a robust neural network based statistical approach, correction of imbalanced data and inter-center variability, a radiomics-based model was externally validated with a balanced accuracy of 85.5%. This model efficiently predicted the patients with a pCR in an external cohort and could be used to select the patients eligible for organ preservation.

**Abstract:**

**Objective**: Our objective was to develop a radiomics model based on magnetic resonance imaging (MRI) and contrast-enhanced computed tomography (CE-CT) to predict pathological complete response (pCR) to neoadjuvant treatment in locally advanced rectal cancer (LARC). **Material:** All patients treated for a LARC with neoadjuvant CRT and subsequent surgery in two separate institutions between 2012 and 2019 were considered. Both pre-CRT pelvic MRI and CE-CT were mandatory for inclusion. The tumor was manually segmented on the T2-weighted and diffusion axial MRI sequences and on CE-CT. In total, 88 radiomic parameters were extracted from each sequence using the Miras© software, with a total of 822 features by patient. The cohort was split into training (Institution 1) and testing (Institution 2) sets. The ComBat and Synthetic Minority Over-sampling Technique (SMOTE) approaches were used to account for inter-institution heterogeneity and imbalanced data, respectively. We selected the most predictive characteristics using Spearman’s rank correlation and the Area Under the ROC Curve (AUC). Five pCR prediction models (clinical, radiomics before and after ComBat, and combined before and after ComBat) were then developed on the training set with a neural network approach and a bootstrap internal validation (*n* = 1000 replications). A cut-off maximizing the model’s performance was defined on the training set. Each model was then evaluated on the testing set using sensitivity, specificity, balanced accuracy (Bacc) with the predefined cut-off. **Results:** Out of the 124 included patients, 14 had pCR (11.3%). After ComBat harmonization, the radiomic and the combined models obtained a Bacc of 68.2% and 85.5%, respectively, while the clinical model and the pre-ComBat combined achieved respective Baccs of 60.0% and 75.5%. **Conclusions:** After correction of inter-site variability and imbalanced data, addition of radiomic features enhances the prediction of pCR after neoadjuvant CRT in LARC.

## 1. Introduction

Colorectal cancer is one of the most common cancers worldwide, with breast and lung cancer, and is particularly frequent in the Western population: in Europe alone, for the year 2020, the estimated incidence of rectal cancer specifically was 113,684 new cases [1]. All stages combined, the five years overall survival reaches approximately 55%, and colorectal represents the second leading cause of cancer death worldwide. The most common histology is adenocarcinoma [2].

For locally advanced rectal cancer (LARC), current guidelines recommend neoadjuvant chemoradiotherapy (CRT) +/− neoadjuvant chemotherapy as proposed by the PRODIGE 23 and RAPIDO trials [3,4] followed by surgery and adjuvant chemotherapy in the case of lymph nodes involvement [5]. Neoadjuvant treatment has demonstrated significant benefits in terms of recurrence-free survival, with complete response rates of 15–20% and up to 28% [3,4], but without any gain in overall survival [6]. Surgery usually consists of a total mesorectal excision (TME). It is now hypothesized that patients with a complete response after neoadjuvant CRT could benefit from a wait-and-see strategy and avoid the morbidity of surgery [6,7] without compromising survival outcomes [8,9,10]. Predicting this complete response to neo-adjuvant treatment is therefore an unmet need in the management of LARC.

Radiomics consists of the extraction of multiple quantitative parameters from medical images. Such features are thought to better apprehend the heterogeneity of the tumor [11,12,13] and have been previously studied in pelvic localizations such as prostate [14], cervical [15], and rectal cancer [16]. A prediction model based on magnetic resonance imaging (MRI) and positron emission tomography (PET) images previously achieved promising results in LARC [17]. However, PET is not currently recommended on a clinical basis yet [18]. On the contrary, contrast-enhanced computed tomography (CE-CT) could be of interest in this context as it is routinely acquired for these patients. Various machine learning tools exist and are constantly being developed. In the specific setting of pCR prediction in LARC, external validation of a model is awaited.

The aim of this study was to build a radiomics model based on pre-therapeutic MRI and CE-CT images to predict pathological complete response (pCR) in patients with LARC treated with neoadjuvant CRT and subsequent surgery and to externally validate this model.

## 2. Materials and Methods

### 2.1. Patient Population

All patients who received neoadjuvant CRT followed by surgery for a LARC from May 2012 to October 2019 in Nantes (Institution 1) and the West Brittany Oncology Institute (Institution 2) were retrospectively screened. Only patients with available MRI with T2 and diffusion sequences, CE-CT, and definitive anatomopathological analysis were included in this study. Patients who did not undergo surgery after neoadjuvant treatment or with incomplete neoadjuvant treatment were excluded.

### 2.2. Outcome

The primary endpoint was the prediction of pCR to neo-adjuvant CRT defined as the absence of tumor residue on the surgical specimen, i.e., ypT0 ypN0.

### 2.3. MRI

The acquisition machines and protocols differed among institutions. Pelvic MRI were performed on two different Siemens 1.T (Siemens Healthcare, Malvern, PA, USA) and a Philips 3T (Philips Healthcare, Eindhoven, The Netherlands), and a Siemens 3T (Siemens Healthcare, Malvern, PA, USA) in institution 1, a Philips 1.5T (Philips Healthcare, Eindhoven, The Netherlands) and 2 Siemens 1.5T (Siemens Healthcare, Malvern, PA, USA) in institution 2, respectively. The images were performed in supine positioning. MRI sequences included axial T2-weighted sequences and axial diffusion sequences using different b-values up to 1000 s/mm^2^. MRI characteristics are summarized in Appendix A.

### 2.4. Contrast-Enhanced CT Scan

For the majority of patients, the radiotherapy (RT) planning CT, if performed with contrast agent injection, was used. In the other cases, the CE-CT performed as part of the initial extension assessment of the disease was collected. CTs were performed in the supine position. Four different CT scans were used including a Siemens and a Philips in institution 1 and two different Siemens in institution 2. The average slice thickness was 2 mm (range 1–5 mm).

### 2.5. Clinical Features

We retrospectively collected the following clinical parameters from each patient’s medical record: initial tumor stage (according to clinical examination, ultrasound-endoscopy, and MRI), initial nodal stage (defined using ultrasound-endoscopy and MRI), carcinoembryonic antigen (CEA) and carbohydrate (CA) 19-9 values, tumor grade on biopsy, distance to anal margin, tumor size, and response to neoadjuvant CRT according to the ypTNM classification.

### 2.6. Tumor Delineation

Rectal tumors were manually delineated in 3D on the different pre-therapeutic sequences, i.e., T2-weighted, diffusion, and on the contrast-enhanced CT separately, by a single expert (A.B.), using the 3D Slicer v4.10.1 software. All segmentations were performed blinded to the pathological response status. Appendix A shows an example of tumor segmentation. In order to assess the robustness of each model to segmentation variability, 25 randomly selected patients from the testing set were segmented by a second expert (V.B), blinded to the first segmentation and the pathological response status (see Section 2.10).

### 2.7. Radiomic Features

Radiomic features were extracted using the Miras© software (LaTIM UMR 1101, Brest, France), compliant with the most up-to-date Image Biomarker Standardization Initiative (IBSI) guidelines and benchmark values [19]. In total, 88 features (15 shape, 11 intensity, and 62 textural features) were extracted from each image (T2, diffusion, and CE-CT). The 62 textural features were calculated with two intensity discretization schemes: linear or equalization (not standardized yet by the IBSI). For both discretization schemes, two fixed bin numbers (FBN), 32 and 64, were used. Texture matrices were built following the merging strategy, i.e., a single matrix taking into account all 13 directions, and a distance of 1 voxel. As a result, 822 features were extracted for each patient from the 3 imaging modalities.

### 2.8. Harmonization Method

We applied the ComBat statistical method using the “COMBAT” package available in R. This a posteriori statistical method aims to account for the imaging modalities’ heterogeneity [20,21] and proved successful in removing intersite technical variability while preserving intersite biological variability. Harmonization was thus performed before feature set reduction. Harmonized features were then processed through the further-defined statistical workflow, developing two new harmonized prediction models: ComBat-Radiomic and ComBat-Combined.

### 2.9. Statistical Analysis

Patients from institution 1 were included in the training cohort while patients from institution 2 formed the testing cohort. Clinical and radiomic features (either original or harmonized) were then selected in the training set only. The feature set reduction workflow was developed as a two-step approach. The first step selected features based on their predictive value of the pCR status. An area under the curve (AUC) ≥ 0.60 was chosen to retain potential features. The second step selected features based on their inter-correlation using the Pearson’s rank correlation coefficient (ρ): when two features appeared as highly correlated (*ρ* > 0.7), only the feature with the highest AUC was kept. The features selected by this two-step approach were then fed to a neural network classifier, using the “Multilayer Perceptron” toolbox available in SPSS Modeler v26.0. Internal validation was performed using bootstrap, with *n* = 1000 replications, for robustness optimization. Before each to building of each model, correction of imbalanced data was performed using the Synthetic Minority Over-sampling Technique (SMOTE). Five models were built in a decremental manner using the clinical, radiomic, and the clinical + radiomics features defining the Clinical, No-ComBat and ComBat-Radiomic, and No-ComBat and ComBat-Combined models, respectively. An optimal cut-off was determined maximizing the Youden Index (YI = Sensitivity + Specificity − 1). Each model’s performance was also evaluated using the AUC and the Receiver Operative Characteristics (Specificity: Sp; Sensitivity: Se; cross-validated Balanced accuracy: Bacc). Each model was then evaluated on the testing set, using sensitivity, specificity, and Bacc.

Statistical analysis was performed using SPSS Modeler v26.0 and MedCalc v15.8.

### 2.10. Inter-Individual Variability

Variability between the initial segmentation and the segmentation performed by the second expert for the 25 randomly selected patients in the testing set was determined using the Dice coefficient. Radiomic features were extracted from these 25 new delineations. Each pre-trained model was then evaluated on this sub-set of patients using these newly extracted features and prediction performances were compared.

### 2.11. Radiomics Quality Score

To evaluate the quality of our study, the Radiomics Quality score was calculated [22].

### 2.12. Ethical Considerations

This study was approved by the hospital ethical committee and all patients gave consent for using their medical data (NCT B2020CE.12). All authors had access to the study data and approved the final manuscript.

## 3. Results

In total, from May 2012 to October 2019, 124 patients were included (64 in institution 1 and 60 in institution 2) with a 1:2 sex ratio (38% women and 62% men) and a median age of 65 years (29–86 years). The flowchart is available in Figure 1. The majority of patients had T2 (13%) or T3 (78.2%) stages and 76% had a lymph node involvement. The initial CEA rate was available for 66% of patients and ranged from 0.8 to 91 ng/mL. Data regarding the histological subtype was available for 27 patients of institution 2. Among these patients, only one patient presented with a mucinous tumor. All patients’ characteristics are available in Table 1.

All patients received neoadjuvant treatment. In total, 118 patients (95%) received CRT, and 6 patients (5%) received neoadjuvant RT only due to contraindications to concomitant chemotherapy. Regarding RT, all patients received a total dose of 45 Gy in 25 fractions to the pelvis with a boost to the rectal tumor up to 50.4 Gy (in 28 fractions) in 44% of them (54 patients). Regarding chemotherapy, 97 patients (78%) received oral Capecitabine and 21 patients (18%) received FOLFOX (5-Fluororacil combined with Oxaliplatin). The mean time between completion of CRT and surgery was 58 days (SD: 13.25). Details of the treatment distribution in the different sets (training and testing) are given in Table 1.

In the whole cohort, 14 patients (11.3%) achieved pCR after neoadjuvant CRT. pCR was equally distributed in the training (*n* = 9, 14%) and the testing (*n* = 5, 8%) sets. The clinical features that stand out as significant predictors of pCR were the initial T-stage and the tumor grade. The clinical model derived from these parameters obtained an AUC of 0.77 (*p* = 0.001) and a Bacc of 65.5% (with a threshold of 8.0% based on the Youden index) in the training set. In the testing set, the clinical model achieved a Bacc of 60.0% with 88.0% of false positives among patients classified at high chance of pCR.

After selection, only 3 features from the T2 sequence (elongation, energy_histogram and entropy_histogram) and 2 from the diffusion sequence (elongation and HGLRE_align; HGLRE: High Gray Level Run Emphasis) remained. None of the CE-CT features were selected. Regarding the training cohort, the No-ComBat Radiomics and No-ComBat Combined models resulted in AUCs of 1.00 and 0.97 and Baccs of 98.2% and 93.6%, respectively. Using their respective cut-offs, each model resulted in a Bacc of 50.9% and 70.0%. No radiomic features extracted from the CT scan were retained in the radiomics and combined prediction models. Considering the radiomic features harmonized with ComBat as input to the feature selection workflow, additional features ended up being retained: from the T2 sequence: mean_histogram, variance_histogram, standard_deviation_histogram, energy_histogram, Gray Level Non Uniformity (GLNU)_norm_align, and from the diffusion sequence: High Gray Level Zone Emphasis (HGLZE) and elongation. However, even after harmonization, no radiomic features extracted from the CE-CT were retained. The most efficient models combining either harmonized radiomic features only or both clinical and harmonized radiomic features achieved respective AUCs/Baccs of 0.62/60.0% and 0.81/85.5%. ROC curves are available in Figure 2a,b for the training and testing cohorts, respectively, while detailed composition of each model and importance of each feature are proposed in Appendix A. Detailed results of each model in the training and testing cohorts are, respectively, available in Table 2 and Table 3.

As shown in Figure 3, the best calibrated models appeared to be the combined model but only after ComBat harmonization.

The analysis of inter-observer variability showed a moderate variability in delineations, with a median DICE coefficient of 0.74 (range 0.41–0.99). The application of the combined prediction model with the ComBat harmonization method provided satisfactory results with an AUC of 0.82 (*p* = 0.002) and a sensitivity of 80.0%, a specificity of 90.0%, and a Bacc of 85.0% with the 6.0% cut-off.

Our study scored 22 points out of 36.

## 4. Discussion

In this study, we evaluated several pCR prediction models in patients treated with neoadjuvant CRT for a LARC. The combined model using harmonized radiomics features performed better than the clinical and radiomics models, with Bacc of 85.5% and the second lowest rate of false positive (55.6%) after the ComBat harmonized radiomics model. This a posteriori ComBat “harmonization’ method certainly enhanced the performance of the radiomics-based models. Depending on the clinician preference, one could either choose the most performant model based on the overall results (ComBat-Combined model) or the ComBat-Radiomics model as it harbors a more favorable profile regarding the risk of false positives.

To our knowledge, the value of the CE-CT scan has never been evaluated in regard to pCR prediction. In our study, we found no added value of the CE-CT-scan for the prediction of pCR when compared to MRI scans. The majority of CE-CT-based radiomics features were highly correlated with the MRI features and were less effective in predicting response to treatment.

Other studies have focused on this subject, using different approaches [16,23]. Shaish et al. [24] considered T2-weighted sequences of the pre-therapeutic MRI in order to predict pCR. Their model resulted in an AUC of 0.66 in the testing population. In a monocentric study, Cui et al. [25] analyzed the value of the multimodality (T1, T2, and ADC sequences), finding an AUC of 0.73 for the T2-based radiomics model and 0.94 for the multimodal-based model. Although the authors showed interest in combining different sequences, their model is subjected to a significant risk of instability and over-fitting due to the large number of factors (12), and has not been externally validated. Zhou et al. [26] developed an efficient model to predict relapse-free survival, which resulted in an AUC of 0.77 in the validation set.

Our objective was to predict pCR based on the pre-CRT clinical and imaging data. Prediction of pCR based on the post-CRT has also been studied in the literature. Liu et al. [27] showed that combination of radiomics signatures from pre- and post-CRT imaging with tumor size could efficiently select patients with pCR, with an AUC of 0.98. In this study, imaging was performed on the same MRI machine, making the analyses homogeneous. However, the variability due to the acquisition on different machines with different parameters was not considered, and therefore, applying this model to data acquired on other devices may be challenging. In addition, it would have been interesting to take into account the variations in radiomic parameters between the pre- and post-CRT MRIs as well as clinical parameters. Horvat et al. [28] looked at the T2-weighted sequences of post-therapeutic MRIs and developed a very efficient model with an AUC of 0.92, but without internal or external validation, and as such, a risk of over-fitting exists. Such approaches would make it possible to provide further arguments in favor of a response or non-response to treatment, while remaining non-invasive.

Genomics has also been studied in this context [29,30]. We previously showed that combining CE-CT radiomics with gene expression analysis and histopathological examination of primary colorectal cancer could provide higher prognostic stratification power [31]. Park et al. [32] specifically reported different genomic signatures in patients with LARC depending on CRT response. Thus, a radio-genomics approach could further enhance the prediction’s performance and offer an optimized and tailored treatment of rectal cancer.

The management of LARC is evolving rapidly. The PRODIGE 23 trial recently reported a trend towards improved disease-free survival, with a significant difference at 3 years, as well as a higher complete response rate with the neo-adjuvant folfirinox chemotherapy sequence followed by pre-operative CRT and total mesorectal excision compared to the standard approach [4]. The RAPIDO trial [3] randomized patients between the short radiation schedule of five fractions of 5 Gy followed by chemotherapy (capox or folfox) compared to the standard of care (CRT with 50.4 Gy) before surgery. A significant decrease in treatment failure and a 2-fold increased response rate compared to standard treatment was found, probably due to the neoadjuvant chemotherapy. As a consequence, therapeutic sequences might change in the near future. Testing our algorithm in this new era will be of particular interest. Surgical management is also on the break of change, especially in the geriatric population where total mesorectal excision could be avoided in favor of transanal endoscopic resection [33]. Going further, several studies have analyzed the possibility of an organ preservation strategy in patients with complete response, with very encouraging results [8,9,10]. It is therefore crucial to be able to properly select for therapeutic strategy adaptation in order to improve quality of life without compromising survival.

The strong point of our study is its multicentric and multiscanner nature. To our knowledge, it is the first external validation of such a model. We also studied the inter-individual variability during segmentation and showed the robustness of the proposed model. Indeed, despite a moderate variability of inter-reader segmentation, the ComBat-corrected combined model achieved similar results when compared to the initial testing set, supporting its robustness. Such stability can be partly explained by the feature selection workflow, the neural network’s approach, and the internal validation process (bootstrap).

Some limitations in our study must be acknowledged. The loss between the training and testing test leaves room for over-fitting correction. Such a task remains troublesome given the low events rate (only 5 pCR patients out of 60 in the testing set), even with the use of the SMOTE correction for imbalanced data. Acknowledging the great heterogeneity of MRI scans in our study, we enhanced the radiomics and combined prediction models after a posteriori statistical harmonization using the ComBat harmonization technique. Our results show the need to use these methods to improve the performance of the model. Finally, the pCR rate of 11.3% found in our study was particularly low compared to other available data with rates close to 20% [5]. This can be explained by a large inclusion period, which started in 2012. Indeed, in 2012, MRIs were used less often in the extension disease assessment, resulting in a possible selection bias. Inclusion of mutational status (RAS/BRAF/MSI) could also partly explain this low pCR rate. Despite this lower percentage, our model was able to predict pCR to neoadjuvant CRT.

## 5. Conclusions

Our radiomic model based on MRI parameters appears to add significant value to usual clinical features for the prediction of pCR after neoadjuvant CRT in patients treated for a LARC while being robust to segmentation variability. Imaging harmonization allowed to increase the prediction performance of our model. Given these promising results, we plan to confirm our findings on a prospective cohort, and to integrate other tumor-specific parameters, notably genomics.

## Figures and Tables

**Figure 1 cancers-14-01079-f001:**
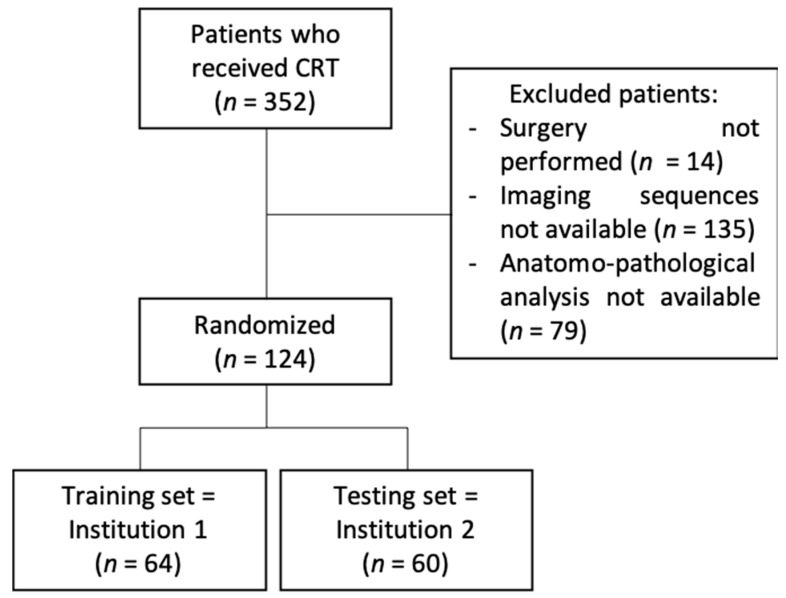
Flowchart.

**Figure 2 cancers-14-01079-f002:**
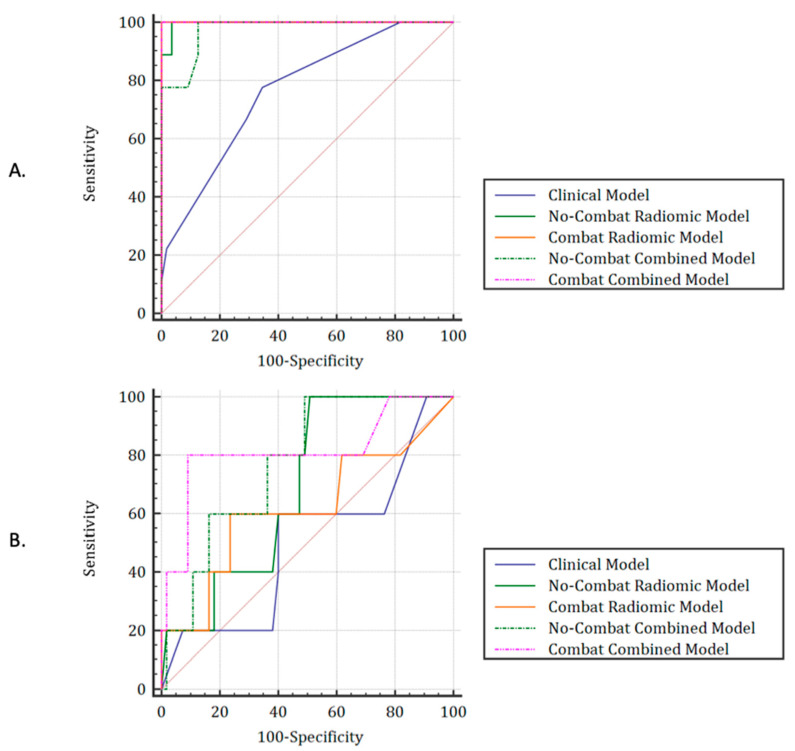
ROC curves for each model in the training (**A**) and testing sets (**B**).

**Figure 3 cancers-14-01079-f003:**
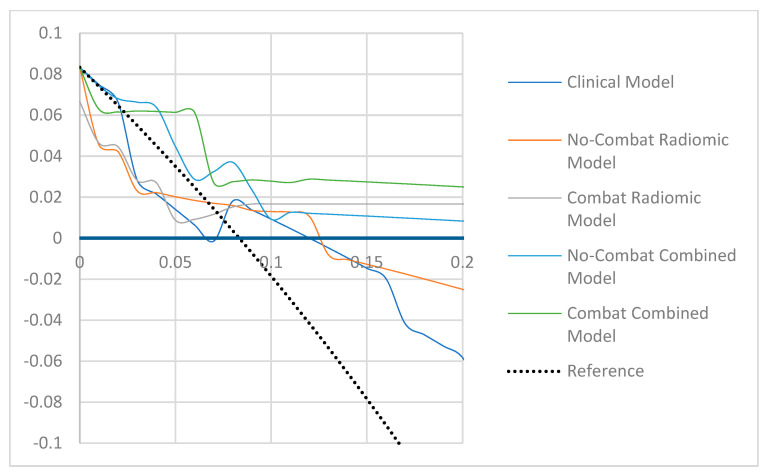
Decisional curve analysis for each model in the testing cohort.

**Table 1 cancers-14-01079-t001:** Initial characteristics.

Variable	Total Cohort *n* = 124	Training Set *n* = 64	Testing Set *n* = 60	*p*-Value
Mean age at diagnosis (years)	65 (SD: 10.75)	62 (SD: 11.8)	68 (SD: 8.4)	0.65
Gender (male/female)	76/47	37/27	40/20	0.91
Degree of differentiation				
Well differentiated (%)	43 (35%)	26 (40.6%)	19 (31.7%)	0.82
Moderately differentiated (%)	58 (47%)	32 (50%)	23 (38.3%)	0.43
Undifferentiated (%)	15 (12%)	1 (1.5%)	14 (23.3%)	0.59
High-grade dysplasia (%)	8 (6%)	4 (6.3%)	6 (10%)	0.59
Mean ACE rate (ng/mL)	8 (SD: 12.27)	6.8 (SD: 7.2)	9.7 (SD: 16.8)	0.80
cT stage				
cT 1 (%)	1 (0.8%)	0 (0%)	1 (1.6%)	0.99
cT 2 (%)	16 (13%)	7 (10.9%)	9 (15%)	0.96
cT 3 (%)	97 (78.2%)	52 (81.3%)	28 (46.7%)	0.23
cT 4 (%)	10 (8%)	5 (7.8%)	4 (6.6%)	0.82
N+ (%)	95 (76%)	50 (78%)	44 (73%)	0.99
pCR (%)	14 (11%)	9 (14%)	5 (8%)	0.75
Radiotherapy	124 (100%)			
3D-CRT	70 (56.5%)	53 (82.8%)	17 (28.3%)	<0.0001
IMRT	54 (43.5%)	11 (17.2%)	43 (71.7%)	
45 Gy to the pelvis only	70 (56%)	59 (92%)	10 (16.7%)	0.04
45 Gy to the pelvis + boost up to 50.4 Gy to the rectal tumor	54 (44%)	5 (8%)	50 (83.3%)	0.03
Concomitant chemotherapy	118 (95%)	64 (100%)	54 (90%)	0.39
Capecitabine	97 (78%)	56 (88%)	41 (68%)	0.37
Folfox	21 (17%)	8 (12%)	13 (21.7%)	0.42
Duration of neoadjuvant therapy (mean, days)	39 (SD: 4.71)	38 (SD: 4.67)	39 (SD: 6.11)	0.93
Delay between the end of treatment and surgery (mean, days)	58 (SD: 13.19)	59 (SD: 12.08)	56 (SD: 15.05)	0.82

**Table 2 cancers-14-01079-t002:** Results of each model in the training cohort (institution 1).

Model	AUC	*p*	Cut-Off (%)	Se (%)	Sp (%)	Bacc (%)	Below the Cut-Off			Above the Cut-Off		
							Total (*n*, %)	TN (*n*, %)	FN (*n*, %)	Total (*n*, %)	FP (*n*, %)	TP (*n*, %)
Clinical	0.77	0.001	8.0	71.2	77.8	65.5	38 (59.4)	36 (94.7)	2 (5.3)	26 (40.6)	19 (73.1)	7 (26.9)
Radiomic	1.00	<0.0001	23.0	100.0	96.4	98.2	53 (82.8)	53 (100.0)	0 (0.0)	11 (17.2)	2 (18.2)	9 (81.8)
Combined	0.97	<0.0001	5.0	100.0	87.3	93.6	48 (75.0)	48 (100.0)	0 (0.0)	16 (25.0)	7 (43.7)	9 (56.2)
ComBat_Radiomic	1.00	<0.0001	17	100.0	100.0	100.0	55 (85.9)	55 (100.0)	0 (0.0)	9 (14.1)	0 (0.0)	9 (100.0)
ComBat_Combined	0.95	<0.0001	6.0	100.0	80.0	90.0	44 (68.7)	44 (100.0)	0 (0.0)	20 (31.2)	11 (55.0)	9 (45.0)

Abbreviations: AUC: area under the curve; Se: sensitivity; Sp: specificity; Bacc: balanced accuracy; TN: true negative; FN: false negative; FP: false positive; TP: true positive.

**Table 3 cancers-14-01079-t003:** Results of each model in the testing cohort (institution 2).

Model	AUC	*p*	Cut-Off (%)	Se (%)	Sp (%)	Bacc (%)	Below the Cut-Off			Above the Cut-Off		
							Total (*n*, %)	TN (*n*, %)	FN (*n*, %)	Total (*n*, %)	FP (*n*, %)	TP (*n*, %)
Clinical	0.50	1.00	8.0	60.0	60.0	60.0	35 (58.3)	33 (94.3)	2 (5.7)	25 (41.7)	22 (88.0)	3 (12.0)
Radiomic	0.69	0.07	23.0	20.0	81.8	50.9	49 (81.7)	45 (91.8)	4 (8.2)	11 (18.3)	10 (90.9)	1 (9.1)
Combined	0.77	0.004	5.0	80.0	60.0	70.0	34 (56.7)	33 (91.1)	1 (2.9)	26 (43.3)	22 (84.6)	4 (15.4)
ComBat_Radiomic	0.62	0.49	17	20.0	100.0	60.0	59 (98.3)	55 (93.2)	4 (6.8)	1 (1.7)	0 (0.0)	1 (100.0)
ComBat_Combined	0.81	0.03	6.0	80.0	90.9	85.5	51 (85.0)	50 (98.0)	1 (2.0)	9 (15.0)	5 (55.6)	4 (44.4)

Abbreviations: AUC: area under the curve; Se: sensitivity; Sp: specificity; Bacc: balanced accuracy; TN: true negative; FN: false negative; FP: false positive; TP: true positive.

## Data Availability

The data presented in this study are available on request from the corresponding author.

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
