# Peer review of "External Validation of a Radiomics Model for the Prediction of Complete Response to Neoadjuvant Chemoradiotherapy in Rectal Cancer"

_cancers, 2022, doi:10.3390/cancers14041079_

Round 1
Reviewer 1 Report
Dear Editor,
thank you for considering me as a reviewer for this paper.
It's a nice written paper, clearly outlined. The english language used is good.
The objective of the present paper was to develop a radiomics model based on magnetic resonance 20 imaging and contrast-enhanced computed tomography to predict pathological complete response to neoadjuvant treatment in locally advanced rectal cancer.
The only minor thing I can comment on (but not mandatory) is to better specify the radiotherapy technique. Were all patients treated with the same technique? VMAT? 3D? were there differences in response?
Please, see line 142 and 143…. Patients from Institution 1 were included in the training cohort while patients from 2 formed the training cohort
Author Response
Dear Editor,
Thank you for considering our work for publication in Cancers.
We would like to thank the reviewers for their comments that helped us further enhancing the quality of our manuscript, entitled “External validation of a radiomics model for the prediction of complete response to neoadjuvant chemoradiotherapy in rectal cancer”. You’ll find below the point-by-point response to the reviewers’ comments. Modifications pertaining to these comments are tracked in the revised manuscript.
Reviewer #1
The only minor thing I can comment on (but not mandatory) is to better specify the radiotherapy technique. Were all patients treated with the same technique? VMAT? 3D? were there differences in response?
Following the reviewer’s suggestion, details regarding the radiotherapy delivery were added in Table 1. As presented, the distribution of the treatment modality was unbalanced between the training and testing sets with no impact on the pCR rate between the two cohorts.
Please, see line 142 and 143…. Patients from Institution 1 were included in the training cohort while patients from 2 formed the training cohort
The manuscript was modified accordingly

Reviewer 2 Report
Dear authors,
I've read your article regarding the use of radiomic for the prediction of pCR after CHRT in rectal cancer with great interest. I think that this a hot tissue currently and the question is of capital importance because it can change treatment for these patients.
Globally I think the paper is well-written and easy to understand. I will only suggest to add some information.
1.- Because of the impact that molecular biology will have in clinical outcomes I strongly recommend adding RAS/BRAF and MSI status for both cohorts (training and testing sets).
2.- Besides, I would recommend adding -if available- the grade of mucinous of the tumors. It has been demonstrated that this histological feature might impact oncological treatments.
Well done.
Author Response
Dear Editor,
Thank you for considering our work for publication in Cancers.
We would like to thank the reviewers for their comments that helped us further enhancing the quality of our manuscript, entitled “External validation of a radiomics model for the prediction of complete response to neoadjuvant chemoradiotherapy in rectal cancer”. You’ll find below the point-by-point response to the reviewers’ comments. Modifications pertaining to these comments are tracked in the revised manuscript.
Reviewer #2
1.- Because of the impact that molecular biology will have in clinical outcomes I strongly recommend adding RAS/BRAF and MSI status for both cohorts (training and testing sets).
As suggested by the reviewer, RAS/BRAF and MSI status are, indeed, critical in clinical outcomes such as progression-free survival and overall survival. However, role of these features for the response to neoadjuvant chemoradiotherapy remains debatable. While the role of RAS/BRAF1 seems to be minor, the impact of the KRAS2,3/MSI is less clear with confounding results. Moreover, mutational status is often not available at the non-metastatic stages of the disease.
In our cohort, data regarding the mutation status was available in only a subset of patients of the Institution 2. For the MSI status, data was available for 13 patients only while informations regarding the RAS/BRAF status were only available for 8 patients.
Lack of these informations was added as a limitation in the discussion paragraph.
-
Oshiro T, Uehara K, Aiba T, Mukai T, Ebata T, Nagino M. Impact of RAS/BRAF mutation status in locally advanced rectal cancer treated with preoperative chemotherapy. Int J Clin Oncol. 2018 Aug;23(4):681-688. doi: 10.1007/s10147-018-1253-z. Epub 2018 Feb 24. PMID: 29478127.
-
Zhou P, Goffredo P, Ginader T, Thompson D, Hrabe J, Gribovskaja-Rupp I, Kapadia M, Hassan I. Impact of KRAS status on tumor response and survival after neoadjuvant treatment of locally advanced rectal cancer. J Surg Oncol. 2021 Jan;123(1):278-285. doi: 10.1002/jso.26244. Epub 2020 Oct 6. PMID: 33022750.
-
Chow OS, Kuk D, Keskin M, Smith JJ, Camacho N, Pelossof R, Chen CT, Chen Z, Avila K, Weiser MR, Berger MF, Patil S, Bergsland E, Garcia-Aguilar J. KRAS and Combined KRAS/TP53 Mutations in Locally Advanced Rectal Cancer are Independently Associated with Decreased Response to Neoadjuvant Therapy. Ann Surg Oncol. 2016 Aug;23(8):2548-55. doi: 10.1245/s10434-016-5205-4. Epub 2016 Mar 28. PMID: 27020587; PMCID: PMC5047012.
-
Hasan S, Renz P, Wegner RE, Finley G, Raj M, Monga D, McCormick J, Kirichenko A. Microsatellite Instability (MSI) as an Independent Predictor of Pathologic Complete Response (PCR) in Locally Advanced Rectal Cancer: A National Cancer Database (NCDB) Analysis. Ann Surg. 2020 Apr;271(4):716-723. doi: 10.1097/SLA.0000000000003051. PMID: 30216221; PMCID: PMC7418064.
-
O'Connell E, Reynolds IS, McNamara DA, Prehn JHM, Burke JP. Microsatellite instability and response to neoadjuvant chemoradiotherapy in rectal cancer: A systematic review and meta-analysis. Surg Oncol. 2020 Sep;34:57-62. doi: 10.1016/j.suronc.2020.03.009. Epub 2020 Apr 2. PMID: 32891354.
2.- Besides, I would recommend adding -if available- the grade of mucinous of the tumors. It has been demonstrated that this histological feature might impact oncological treatments.
In our cohort, data regarding the histological subtype was available in only a subset of patients of the Institution 2 (27 patients). Among these 27 patients, only one patient presented with a mucinous tumour. This information was added in the results paragraph.
